# Transcriptome Analysis of Compensatory Growth and Meat Quality Alteration after Varied Restricted Feeding Conditions in Beef Cattle

**DOI:** 10.3390/ijms25052704

**Published:** 2024-02-26

**Authors:** Tianyu Deng, Mang Liang, Lili Du, Keanning Li, Jinnan Li, Li Qian, Qingqing Xue, Shiyuan Qiu, Lingyang Xu, Lupei Zhang, Xue Gao, Junya Li, Xianyong Lan, Huijiang Gao

**Affiliations:** 1Institute of Animal Sciences, Chinese Academy of Agricultural Sciences, Beijing 100193, China; dty9527@126.com (T.D.); liangmang87@163.com (M.L.); dulili1996@126.com (L.D.); likeanning@163.com (K.L.); ljn20982906@163.com (J.L.); pbli0201@163.com (L.Q.); xqq18292947845@126.com (Q.X.); qsy527518520@foxmail.com (S.Q.); xulingyang@163.com (L.X.); zhanglupei@caas.cn (L.Z.); gaoxue76@126.com (X.G.); lijunya@caas.cn (J.L.); 2Shaanxi Key Laboratory of Molecular Biology for Agriculture, College of Animal Science and Technology, Northwest A&F University, Yangling 712100, China

**Keywords:** compensatory growth, transcriptome, cattle, meat quality

## Abstract

Compensatory growth (CG) is a physiological response that accelerates growth following a period of nutrient limitation, with the potential to improve growth efficiency and meat quality in cattle. However, the underlying molecular mechanisms remain poorly understood. In this study, 60 Huaxi cattle were divided into one ad libitum feeding (ALF) group and two restricted feeding groups (75% restricted, RF75; 50% restricted, RF50) undergoing a short-term restriction period followed by evaluation of CG. Detailed comparisons of growth performance during the experimental period, as well as carcass and meat quality traits, were conducted, complemented by a comprehensive transcriptome analysis of the *longissimus dorsi* muscle using differential expression analysis, gene set enrichment analysis (GSEA), gene set variation analysis (GSVA), and weighted correlation network analysis (WGCNA). The results showed that irrespective of the restriction degree, the restricted animals exhibited CG, achieving final body weights comparable to the ALF group. Compensating animals showed differences in meat quality traits, such as pH, cooking loss, and fat content, compared to the ALF group. Transcriptomic analysis revealed 57 genes and 31 pathways differentially regulated during CG, covering immune response, acid-lipid metabolism, and protein synthesis. Notably, complement–coagulation–fibrinolytic system synergy was identified as potentially responsible for meat quality optimization in RF75. This study provides novel and valuable genetic insights into the regulatory mechanisms of CG in beef cattle.

## 1. Introduction

Compensatory growth (CG) represents an accelerated growth process occurring when an animal is adequately re-fed following a period of nutrient deficiencies or restriction [1]. CG constitutes a crucial physiological phenomenon within the animal production system, especially significant in cattle production [2]. This growth pattern aids in reducing feed costs and typically enhances the feed efficiency of the animal during the re-feeding period [3]. In addition to leveraging CG for enhancing animal growth efficiency, CG may also impact meat quality characteristics; however, this effect can be complex and varies depending on the experimental design and CG factors [4,5,6].

The intensity and effectiveness of CG are influenced by various factors, encompassing the degree and duration of feed restriction, re-feeding period, as well as the animal’s sex and genotype [7,8,9]. For instance, a short-term and not too severe feed restriction may lead to more effective CG [10]. In addition, the effectiveness of CG is influenced by the animal’s stage of growth, which may have a synergistic effect on CG when puberty occurs concurrently with re-feeding [8]. The physiological and molecular mechanisms of CG after feed restriction have been partially investigated [1,9,11,12]. It was found that during feed restriction, growth hormone (GH) production and secretion were enhanced, but the number of GH receptors was reduced, leading to a decrease in GH resistance and insulin-like growth factor-I (IGF-I) secretion. During re-feeding and CG, insulin secretion was sharply enhanced, and plasma GH concentrations remained high, which may have allowed more nutrients to be utilized for the growth process [1]. In addition, the rates of protein synthesis and degradation during compensatory growth were shown to be accelerated, possibly by regulating transcriptional activity in muscle tissue [10]. In contrast, at the molecular level, key genes during compensatory growth are often involved in energy metabolism, protein synthesis and degradation, and muscle growth and differentiation [11,13]. Nevertheless, based on the complexity of CG, a large number of observations and studies are needed to fill the gaps in the understanding of its molecular regulatory mechanisms. Studies on the molecular mechanisms underlying the effects of CG in beef cattle at different levels of restriction and its impact on beef product quality remain absent.

In this study, we compared the growth performance and carcass and meat quality traits between unrestricted and compensating animals with different feeding restriction levels in the Huaxi cattle followed by comprehensive analyses of key genes involved in CG and their phenotypic regulation utilizing the *longissimus dorsi* muscle transcriptome data in an integrated manner. The findings of this study enrich the understanding of the regulatory mechanisms of CG metabolism, broaden our knowledge of CG in beef cattle, and contribute to genome-assisted breeding programs aimed at selecting animals with greater CG capacity and superior meat quality.

## 2. Results

### 2.1. Animal Performance of Growth and Meat

Various growth traits were assessed across different feeding treatment groups throughout the experimental period. The corresponding phenotypic data are summarized in Table 1. Briefly, all three cattle groups, initially similar in body weight (BW), exhibited significant differences (*p* < 0.001) at the end of the restricted feeding phase (d49); RF75 and RF50 were 46 and 84 kg lighter than ALF on average, with RF50 even weighing less than at the experiment’s start. The BW difference diminished progressively during the nutritional recovery period and vanished at slaughter; by day 86 (d135) of realimentation, the BW of RF75 was comparable to ALF in multiple assessments, with final body weights ranking as RF75 (774.05 kg) > ALF (767.18 kg) > RF50 (759.9 kg). In the restricted feeding phase, the ALF group exhibited the highest weight gain followed by RF75 and then RF50. During the initial phase of realimentation (d49–d135), both RF75 and RF50 exhibited marked CG, achieving average daily weight gains (ADG) of 1.47 kg/day and 1.51 kg/day, respectively. Subsequently, although their growth rates decreased, they remained superior to those for the ALF group, with the entire realimentation period’s ADG being 1.12 kg/day for RF75 and 1.15 kg/day for RF50, notably outpacing ALF’s 0.93 kg/day. The rapid catch-up of the two restricted feeding groups caused a nearly equal growth performance with that of the ALF group during the whole experiment.

Restricted feeding and full compensatory growth led to similar values in carcass weight, eye muscle area, and yield of several important meat cuts including striploin, high-rib, and tenderloin across the three groups (Table 2). Conversely, the feeding treatment exhibited a notable influence on certain meat quality and compositional traits. Notably, at least one restricted feeding group showed significant differences from the ALF group in 24-h post-mortem pH (pH_24_), cooking loss, and proportions of protein and fat content. Specifically, the average pH_24_ in RF75 and RF50 was significantly higher than ALF by 0.35 (*p* < 0.01) and 0.20 (*p* < 0.05), respectively. RF50 had the worst cooking loss value at 0.24, which was significantly lower than other two groups with no significant differences. Protein content in the meat of the RF75 group was significantly lower compared to the ALF group, with RF50 showing intermediate protein levels. Correspondingly, fat content followed an inverse pattern to protein content, with RF75 having a 0.82% higher fat content than the ALF.

### 2.2. Evaluation of RNA-Seq Data

In this study, 49 cattle (16 ALF, 17 RF75, and 16 RF50) were finally selected for transcriptome analysis based on sample and sequencing quality assessment (Appendix A). After sequencing the *longissimus dorsi* muscle tissue, we obtained an average of 44.7 million raw reads per sample. After quality control, the average Q20 score and Q30 score of the samples were 97.8% and 93.1%, respectively, reflecting the high fidelity of the sequencing process. Subsequent alignment to the reference genome showed an average mapping rate of 96.5%, with 89% of paired reads uniquely aligned, ensuring efficient downstream differential expression analysis.

### 2.3. Identification of DEGs in Longissimus Dorsi Muscle

In our study, principal component analysis (PCA) was first performed to visualize the overall expression patterns among the three groups. The PCA results (Figure 1A) showed that the ALF, RF75, and RF50 groups formed distinct clusters with clear differences. Meanwhile, the heatmap of the top 1000 genes with the highest variance across samples revealed different expression profiles, clearly separating the three groups (Figure 1B). With the criteria *p*-adj < 0.05 and |log_2_FoldChange| > log_2_1.5, differential expression analysis between each pair of groups identified a total of 1170, 595, and 541 DEGs for RF75 vs. ALF (Appendix A), RF50 vs. ALF (Appendix A), and RF75 vs. RF50 (Appendix A), respectively. Six overlapping genes were identified across the three comparisons: *TNFRSF12A*, *SRXN1*, *IL1RAP*, *IQCA1L*, *PITX3* and one uncharacterized gene ENSBTAG00000001143 (Figure 1C). Furthermore, as shown in Figure 1D, the number of up-regulated genes was 803 for RF75 vs. ALF, 427 for RF50 vs. ALF, and 334 for RF75 vs. RF50 comparisons, constituting over 61% of the total DEGs in each comparison. This predominance of up-regulated genes indicates a pronounced increase in gene expression under CG. Volcano plots in Figure 1E further delineated the distribution of DEGs, highlighting the magnitude and direction of expression changes for both up- and down-regulated genes.

### 2.4. Functional Enrichment Analysis of DEGs

To understand the putative functions of the DEGs, Gene Ontology (GO) enrichment analysis, utilizing our custom-built GO database, revealed that each comparison was significantly enriched in unique biological pathways. Specifically, the RF75 vs. ALF comparison was enriched in 631 pathways (Appendix A), RF50 vs. ALF in 585 pathways (Appendix A), and RF75 vs. RF50 in 924 pathways (Appendix A). Overall functional profiling through network analysis of the top 100 pathways across comparisons delineated distinct biological themes associated with varying levels of restricted feeding. For example, 75% feeding restriction predominantly influenced gene expression related to acid, lipid, small molecule, and lipoprotein metabolism functional pathways while activating growth compensation. In contrast, 50% feeding restriction more significantly affected gene expression in pathways pertinent to cell development, adhesion, stimulus response, and immune function. Furthermore, a substantial proportion of pathways enriched in the RF75 vs. RF50 comparison were common to the other comparisons, indicating that the degree of feeding restriction modulates gene expression in a subset of pathways consistently responsive to dietary limitations.

Consistently, focusing on the shared pathways between RF75 vs. ALF and RF50 vs. ALF revealed unique aspects of feeding restriction and compensatory growth (Figure 2B). The distinct pathways across all groups were primarily concentrated in regulation of response to stimulus (GO:0048583), immune system process (GO:0002376), extracellular region (GO:0005576), ion binding (GO:0043167), and molecular function regulator activity (GO:0098772). Additionally, when comparing the RF groups to ALF, there was a marked enrichment in pathways associated with developmental process, MAPK cascade, muscle structure development, cell junctions, cytoskeletal protein binding, and actin binding, indicating dramatic changes in cellular architecture and functionality during growth compensation.

### 2.5. Gene Set Enrichment Analysis (GSEA) and Gene Set Variation Analysis (GSVA)

In the present study, GSEA and GSVA were conducted to elucidate the functional pathways implicated in restricted feeding and compensatory growth. As illustrated in Figure 3, the most representative pathways from each comparison were visualized in GSEA plots. The results indicated that, compared to the ALF group, RF75 exhibited enhanced activity in metabolic enzyme activity, acid-lipid metabolism, and blood coagulation as well as decreased activity in protein folding (Figure 3A, Appendix A). Conversely, RF50 showed a pronounced negative impact on ribosome structure and function and protein synthesis and folding, alongside a comprehensive activation of the immune system, ranging from invader detection to immune cell activation (Figure 3B, Appendix A).

Additionally, the comparison between RF75 and RF50 revealed not only some overlapping pathways from the previous comparisons but also down-regulation of transmembrane signal transduction activity and MAPK cascade regulation, coupled with an up-regulation of ion binding, organic acid metabolism, and triglyceride metabolism (Figure 3C, Appendix A). The integrative upset plot, combining GSEA and GO enrichment analysis, delineated the extensive overlap between RF75 vs. ALF and RF75 vs. RF50 or RF50 vs. ALF and RF75 vs. RF50, while the overlap between RF75 vs. ALF and RF50 vs. ALF were notably limited (Figure 3D).

Consistently, the GSVA revealed that the most significant pathways identified in the three comparisons closely aligned with the results obtained from previous enrichment analyses (such as the amino acid metabolism pathway is active in RF75 vs. ALF and protein folding is diminished in RF50 vs. ALF) (Figure 4A). In total, 160, 107, and 91 significant pathways were identified in the RF75 vs. ALF (Appendix A), RF50 vs. ALF (Appendix A), and RF75 vs. RF50 (Appendix A) comparisons, respectively. Notably, compared to RF75, the RF50 group exhibited a higher number of negatively regulated pathways, a finding corroborated by the correlation heatmap between samples and GSVA scores in Figure 4B. When examining the concordance of pathway significance obtained from GSVA with those derived from GSEA (Figure 4C) and GO (Figure 4D) enrichment methods, it was observed that the correlation of GSVA *p*-values for both methods was not high, particularly in the RF75 vs. RF50 comparison. Additionally, GSVA tended to detect significance in gene sets with fewer genes, representing more finely categorized pathways. These gene sets often correspond to more specific biological processes, highlighting the nuanced differences detected by GSVA.

### 2.6. Weighted Gene Co-Expression Network Analysis

To comprehensively investigate whether there are correlations between restricted feeding and CG with key genes, WGCNA was employed to analyze the differential expression across the three groups. The co-expression networks were constructed using a soft-thresholding power, with the parameter β set to 5 to maintain a scale-free topology, as networks with fit indices greater than 0.85 were considered scale-free (Figure 5A). The adjacency matrix was generated using the adjacency function, and hierarchical clustering was performed using the topological overlap matrix (TOM) as the dissimilarity measure (Figure 5B). Excluding the grey module, 17 co-expression modules were identified. Modules with a significance level of *p* < 0.05 were designated as key modules. Notably, the turquoise (|r| = 0.84, *p* = 5 × 10^−14^), pink (|r| = 0.78, *p* = 4 × 10^−11^), and yellow (|r| = 0.52, *p* = 1 × 10^−4^) modules showed the strongest positive correlations with the ALF, RF75, and RF50 groups, respectively (Figure 5C). Additionally, the green and blue modules were selected as key modules due to their strong correlations. Within these five key modules, 288 genes were identified with |MM| > 0.8 and |GS| > 0.2 (Figure 5D, Appendix A).

### 2.7. Integrated Analysis to Identify Key Genes

In an integrated approach to identify key genes and pathways involved in the biological processes of compensatory growth under study, we combined the results from WGCNA, differential expression analysis, GSEA, GSVA, and pathway enrichment analysis. The Sankey diagram with bubbles provided a visual representation of the flow from WGCNA modules to genes and then to pathways, illustrating the interconnectedness of genes identified by multiple methods and their subsequent mapping to biological pathways (Figure 6A). Specifically, we identified 57 key genes and GO pathways/terms (Appendix A) consistently implicated across all methods as crucial in facilitating compensatory growth following restricted feeding. For example, protein folding-related functions were diminished in the restricted feeding groups of cattle, attributed to altered expression of *HSPE1*, *HSPB1*, *HSPA8*, and *HSP90AA1* within the turquoise module. Conversely, genes from the apolipoprotein family, including *APOA2*, *APOC3*, *APOE*, and *APOH*, along with complement system components *C3*, *C4BPA*, *C8A*, *C9*, and *CFB* from the pink module were found to enhance lipid anabolism and complement activation in the RF75 group. Furthermore, genes like *KLHL40* and *LMOD3* in the blue and green modules were indicative of altered myofiber structure in RF75. In comparison, the RF50 group exhibited more pronounced extracellular matrix (ECM) structure and remodeling, driven by genes such as *COL1A1*, *COL1A2*, and *MMP2* within the yellow module. In summary, these results clearly demonstrate the role of key genes in regulating compensatory growth, affecting protein, lipid, immune, and cellular structural functions.

Protein–protein interaction (PPI) network analysis was performed for 57 key genes using STRING, identifying 48 that formed a robust PPI network after excluding low-confidence interactions and isolated nodes. As shown in Figure 6B, the network separated into four distinct clusters. The functions of the genes pooled in each cluster were aligned with the integration results described above, and the parts were connected by central nodes with high connectivity, such as *C3*, *F2*, *APOE*, *PLG*, *HSPA8*, *MMP2* and *CYP1A2*, highlighting that these genes play pivotal roles in orchestrating complex biological pathways for compensatory growth.

Finally, in an attempt to widely validate the evidence for the genetic basis of the 57 key genes with phenotypic traits, we explored relevant QTLs affecting production traits, such as growth and meat quality, around these key genes using Cattle QTLdb. As a result, 218 QTL within a range of 100 Kb upstream and downstream of 42 genes were found to be significantly associated with 11 production traits, including “residual feed intake”, “body weight”, and “Average daily gain”, and 15 meat and carcass traits, including “fat percent”, “carcass weight”, “marbling score”, and “shear force”. The above findings demonstrated an important genetic link between key genes and changes in growth and development, feed efficiency, muscle composition, and fat deposition efficiency during CG in Huaxi cattle.

## 3. Discussion

This study designed a short-term restricted feeding experiment followed by CG, aiming to observe the comprehensive effects of long-term CG on the final growth, meat quality, and slaughter-related economic traits of beef cattle. Additionally, at the transcriptome level, it compared and analyzed the relevant genes and biological pathways of compensatory growth under different degrees of feed restriction to elucidate the phenotypic variation patterns of animals that have experienced CG. We found that regardless of the degree of feed restriction, animals subjected to restricted feeding were always able to catch up or even surpass the BW of ad libitum fed animals through CG. For this reason, there were no significant differences in the weights of the several most economically valuable meat pieces constituting the carcasses of the animals. Moreover, the beef produced by compensating animals was superior in terms of pH and fat content compared to that from ad libitum fed animals at the cost of reduced protein content. Additionally, the analysis of the muscle transcriptome elucidated the functional mechanisms of growth compensation and also validated the above phenotypic changes. Therefore, by simultaneously considering the degree of feed restriction, phenotypic changes, and transcriptomic molecular mechanisms, we revealed the effects of CG in a comprehensive and systematic manner, thereby expanding our understanding and application of CG in beef cattle.

In general, compensatory growth can be classified into over-compensatory growth, completely compensatory growth, partially compensatory growth and non-compensatory growth, depending on the degree of compensation [14]. The effects of compensatory growth are often influenced by factors such as sex [9,15], genotype [4,7], stage of growth [8], and degree and duration of feed restriction [10]. Considering the challenges in sourcing animals that are homogeneous in terms of origin, growth stage, and genotype, along with the constraints of experimental facilities in feeding and management, we meticulously selected 20 animals per group. This was to strike a balance between scientific rigor and logistical feasibility. While this number was deemed adequate for our comparative genomics analysis, it is our anticipation to address these challenges in future studies. As our capacity to manage larger cohorts improves, we aim to include more animals, thereby enhancing the robustness and reliability of the genetic analysis. In this study, young bulls around 12 months of age were selected as they tend to have high compensatory growth potential. Additionally, a nearly two-month restriction period followed by a ten-month recovery period allowed us to better observe the phenomenon of CG and its effects on the animals. Unlike the findings of Keady et al. [4], in our study, the Huaxi cattle in the RF50 and RF75 restricted feeding groups exhibited CG throughout the realimentation period and eventually achieved completely CG, with the RF75 group even surpassing the ALF group in final weight, although the difference was not yet significant. Conversely, in their studies of Angus × Belgian Blue offspring at the same growth stage, Keady et al. found that CG ceased after 150 days of the realimentation period, preventing the restricted animals from catching up with the ad libitum fed animals in terms of weight. Another study on Holstein Friesian bulls [3] with a shorter recovery period also showed that the restricted animals did not catch up with the weight of the ad libitum fed animals, highlighting the excellent potential CG capacity of Huaxi cattle.

Extensive research has been conducted on the impact of CG on meat quality and its underlying causes; however, the findings are frequently contradictory [4,5,6]. As previously mentioned, our findings indicated that the pH decline in the RF75 group was lower compared to the ALF group, suggesting a lower glycogen breakdown [6], thereby decreasing lactic acid accumulation in muscle post-slaughter in animals undergoing CG and potentially enhancing meat quality. This observation contradicts findings from other studies in beef cattle [4,6], yet aligns with results from a study on CG in pigs [16]. Additionally, regarding the elevated muscle fat content observed in the RF75 group, compared to both ALF and RF50, one plausible explanation is that animals experiencing CG tend to develop more in line with their physiological age rather than their chronological age [17,18]; thus, once the compensating animals catch up with the weight of ad libitum fed animals, they enter the muscle fat deposition stage more rapidly. The reduction in beef protein content in the restricted groups aligns with previous conclusions about CG affecting protein synthesis and degradation rates, and it is assumed that the proteolytic systems involved in muscle accretion also operate post-mortem such that an increase in protein synthesis and in particular protein degradation during rapid growth will be reflected in an increase in post-mortem proteolysis [5,19,20]. Additionally, research on the relative impact of CG at different levels of feed restriction on beef quality in similar production systems is quite limited. In our study, apart from cooking loss, the meat quality of the more severely restricted group (RF50) tended to be between that noted for RF75 and ALF, whereas the differences in cooking losses in CG animals are often due to breed differences [4,6,21]. In Belgian Blue cattle, compensating cattle showed greater cooking losses, but this was not observed in Angus cattle in the same study. Moloney et al. [6] also reported no differences in cooking losses in Friesian castrates. Additionally, the thermal properties of collagen in intramuscular connective tissue (IMCT) are also considered as one of the significant factors influencing cooking loss. IMCT comprises both labile and stable collagen. The former is prone to dissolution during cooking or degradation post-mortem, while the latter exhibits resistance to cooking and post-mortem aging [22,23]. Although thermal degradation of collagen was not detected in our study, the subsequent transcriptomic analysis revealing alterations in the expression levels of collagen-related genes solely in the RF50 group implies this possibility. Those results discussed above collectively indicate that RF75 exhibits relatively superior meat quality performance, suggesting that judicious utilization of CG can enhance meat quality in Huaxi cattle, a muscle-type beef breed. This insight holds significant implications for the beef industry, emphasizing the importance of harnessing CG to improve meat quality characteristics in cattle production. Meanwhile, all these findings underscore that CG is a complex physiological process influenced by multiple factors. To better understand its biological mechanisms, we further explored the gene expression patterns in different groups of *longissimus dorsi* muscle in more detail at the transcriptomic level.

We employed several commonly used transcriptomic analysis methods in our study, including traditional post-transcriptome differential expression enrichment analysis, GSEA, GSVA, and WGCNA. The purpose of this approach was to ensure the universality and accuracy of our results, validating the findings through multiple methods to maintain the rigor of our research. From differential enrichment analysis to GSEA and then to GSVA, we observed that significant pathways tended to be more finely stratified. This is because research methods using gene sets as units of expression [24] are more influenced by the consistency of gene expression levels within the gene sets. In other words, the power of gene set analysis is enhanced only when all differentially expressed genes (DEGs) in the gene set are either up-regulated or down-regulated (which is more likely in more finely stratified pathways) [25]. In this respect, unsupervised competitive testing methods (GSVA) are more sensitive than supervised competitive testing methods (GSEA) [25,26,27,28]. Consequently, this approach enabled us to understand the differences in CG mechanisms under different levels of feed restriction from a macro to a micro perspective in greater detail.

Overall, compared to animals that grow normally with ad libitum feeding, regardless of the degree of feed restriction, compensating animals exhibit changes in biological processes, such as cell development and differentiation, protein folding, stimulus reaction, immune response, and cell communication. It is known that during re-feeding and the CG period, the secretion of insulin sharply increases, and the concentration of growth hormones in the plasma remains high, gradually decreasing after reaching the peak (at about 60 days in cattle) of CG [1]. In the compensating animals, up-regulation was observed in various growth-related genes, such as *IGF1*, *IGF1R*, *IGFBP5*, etc. (Appendix A). However, given that these expression data were derived from the later stages of CG, these genes were not prominently distinguished among the numerous DEGs. Additional differences between the peak and late CG phases were evident in our observation that genes associated with protein folding, particularly those within the heat shock protein (HSP) family, were down-regulated in compensating animals. This group includes *HSP90AA1*, *HSPA8*, *HSPB1*, *HSPE1*, and several DNAJ genes, also known as Hsp40 homologs [29,30]. This observation contradicts several prior studies that reported an up-regulation of certain HSPs during the peak of CG, associated with increased protein synthesis and tissue deposition [13,31,32]. This discrepancy could be attributed to the fact that in the early stages of CG, an increased need for protein folding mechanisms is likely, owing to the substantial increase in protein synthesis. However, as CG advances, cells might adapt to the heightened synthetic load and develop a tolerance for protein synthesis, consequently diminishing the necessity for these HSPs. This tolerance appears to be more pronounced in animals undergoing more severe feed restriction (RF50) and is manifested in the down-regulation of ribosomal function-related pathways. Interestingly, the observed up-regulation of genes related to collagen production and degradation in the extracellular matrix of RF50 animals, such as *COL1A1*, *COL1A2* and *MMP2*, may partly explain why the muscle protein content in the RF50 group was marginally lower than that in the ALF group, yet the cooking loss was significantly higher [23,33,34]. In another study focusing on feed intake and weight gain in beef cattle, several HSPs overlapping with those identified in our research were found to be up-regulated in low-feeding-low-growth cattle [35], suggesting a potential relationship between these HSPs and compensatory animal feed utilization.

Nutrition and immunity are closely intertwined. Nutritional and caloric deficiencies activate the hypothalamus–pituitary–adrenal axis, mobilizing the body’s immune system while regulating growth metabolism [36]. Our results demonstrate that this immune mechanism remains after restoring nutrients. In our comparative study, although both groups of compensating animals activated the immune system, the immune pathways activated varied with different levels of feed restriction. The complement system was more notably activated in the RF75 group compared to the RF50 group. The complement system, an essential part of the innate immune system composed of proteins like C1–C9, is involved in immune responses, inflammatory reactions, phagocytosis, and metabolic pathways [37,38]. After nutritional recovery, the concentrations of free fatty acids (FFA), glucose (Glc), insulin, and chylomicrons in plasma and tissues increase, promoting complement activation, leading to high levels of anaphylatoxins C3a and C5a and C3adesArg (ASP) locally and systemically. In turn, complement components, through binding with C3a and C5a to their receptors C3aR and C5aR, respectively, promote triglyceride (TG) formation by inhibiting fat breakdown, enhancing Glc and FFA absorption, and indirectly reducing FFA release [39,40,41]. Adipose tissue cells also produce various complement factors, such as factor D (adipsin, *CFD*), factor B (*CFB*), factor H (*CFH*), and *C3* [38,42,43]. Additionally, the complement system can influence food intake and energy expenditure by acting on the central nervous system [44]. The complement, coagulation, and fibrinolysis systems have co-evolved in living organisms [45,46,47], often synergizing through their respective pivotal proteins, namely, complement C3 (*C*3), prothrombin (*F*2), and plasminogen (*PLG*), to mediate integrated physiological responses [47,48,49]. The protein–protein interaction network we constructed also reflects the phenomenon. Consequently, we observed that activation of the complement system in the RF75 group was coupled with concurrent activation of both the fibrinolytic and coagulation systems. A majority of the genes identified in these systems demonstrate pleiotropy, possessing fibrinolytic or coagulation functions and concurrently regulating immunity, enzymatic, and lipid metabolism functions. For example, apolipoproteins (including *APOE*, *APOH*, and *APOA*) are crucial in regulating lipid homeostasis in the central nervous system and blood, playing a role in the distribution and redistribution of lipids across diverse tissues and cells in animals [50,51]. *APOE*^−/−^ mice supplemented with phytosterols show reduced cholesterol levels in serum and liver, yet exhibit increased brain cholesterol levels [52]. Intracerebroventricular administration of *APOE* notably reduced food intake in rats, and re-feeding for 4 h post-fasting induced a significant rise in hypothalamic *APOE* mRNA levels, suggesting that *APOE* may be a key factor in controlling appetite [53]. In addition, we also found up-regulation of many metabolism-regulating enzymes and peptide hormone-related genes in RF75, which appeared to be affected by changes in coagulation and fibrinolytic activities. Transthyretin (TTR), a well-known carrier protein, is responsible for the transport of thyroid hormones in animals [54] and is also implicated in cellular processes including protein hydrolysis, neurogenesis, and autophagy [55,56]. The overexpression of *TTR* appears to regulate adipogenesis and differentiation in cattle, mediated through the enhanced expression of peroxisome proliferator-activated receptor γ (PPARγ) and fatty acid binding protein 4 [57]. 11β-Hydroxysteroid dehydrogenase type 1 (*HSD11B1*), 17β-hydroxysteroid dehydrogenase type 6 (*HSD17B6*), and type 11 (*HSD17B11*) are NADPH/NAD^+^-dependent oxidoreductases, catalyzing the oxidation-reduction of steroid substrates primarily responsible for the oxidation-reduction of hormones, fatty acids, and bile acids [58,59]. Significantly, these enzymes also play a role in the regulation of oxidative stress and digestion [60,61]. These findings suggest that the complement–coagulation-fibrinolytic cascade may contribute to fat deposition in the muscle of RF75 animals. However, this process may result in further diminishing of the proteins composing muscle tissue, potentially leading to impaired cellular structural stability, possibly more severe than that noted in RF50.

Severe malnutrition has been shown to impair cytokine responses and affect the migration of immune cells [36,62,63]. RF50 animals exhibited an overactive adaptive immune response, as evidenced by heightened activities in transmembrane signaling transduction, lymphocyte and leukocyte migration, and G protein-coupled receptor activities. These alterations may be intricately linked to the growth potential, health status, and aging processes in compensating animals [63,64]. However, due to the complexity of these interactions and the constraints of this article’s scope, further investigation is warranted.

In summary, our findings reveal the complexity of gene expression related to growth development, immune stress, lipid, and protein metabolism of CG. This provides a new perspective for further research into the principles of CG. Our results suggest that our understanding of this complex physiological process is far from complete, necessitating additional experiments and analyses to elucidate the specific roles and regulatory mechanisms of these genes in animal CG. These findings enrich the understanding of the regulatory mechanisms of CG metabolism, broaden our knowledge of CG in beef cattle, and contribute to genome-assisted breeding programs aimed at selecting animals with greater CG capacity and superior meat quality.

## 4. Materials and Methods

### 4.1. Animal and Experiment Design

In this study, all experimental cattle were sourced from Huaxi cattle resource population established in the Ulgai (Inner Mongolia, China). Following weaning, the calves were transferred to Jingxinxufa Farm (Chengde, China) to undergo uniform feeding. Upon reaching 12 months of age, 60 bulls were selected and randomly assigned to one of three groups (20 per groups), ensuring comparable body condition across the groups. The groups were categorized as follows: the ad libitum feeding group (ALF), the restricted feeding 75% group (RF75), and the restricted feeding 50% group (RF50). The experiment encompassed three distinct phases. The initial phase, the acclimatization period, spanned one week. In this phase, cattle were stanchion fed, with each animal allocated a separate chute. Feed formulation adhered to the Beef Cattle Feeding Standard (NY/T 815-2004 [65]) to satisfy the nutritional needs of the cattle. Throughout this phase, water was available ad libitum, and Appendix A displays the basal composition and nutrient levels of the feeds. The daily feed intake of cattle was recorded during the acclimatization period, and it was ensured that the remaining feed in the troughs was consistently maintained at approximately 5%. The subsequent phase, a seven-week restricted feeding period, ensued. In this phase, the RF75 and RF50 groups consumed 75% and 50%, respectively, of their average feed intake from the acclimatization period. Concurrently, the ALF group continued with ad libitum feeding. Feed intake for the ALF group was continually recorded, and adjustments were made weekly to the RF75 and RF50 groups’ intake based on observed changes. The final phase encompassed a realimentation and growth compensation period, wherein all three cattle groups resumed ad libitum feeding. Upon reaching 24 months of age, the cattle were transported to Zhongao Foods Ltd. (Chifeng, China) for slaughter. Body weight was recorded multiple times throughout the experimental cycle, including on the day before and during the restricted feeding period, throughout the growth compensation period, and before slaughter, using the weight on the day before the restricted feeding period as the baseline (day d0). During slaughter, the *longissimus dorsi* muscle between the 12th and 13th ribs was harvested for meat quality assessments in accordance with the China National Beef Carcass and Cuts Standards (GB/T 27643-2011 [66]), and partial samples were collected. These were immediately frozen using liquid nitrogen and stored at −80 °C pending RNA extraction.

### 4.2. Meta Quality and Composition

#### 4.2.1. pH

The pH of the *longissimus dorsi* samples was assessed using a HANNA HI99163 Meat pH meter (Hanna Instruments Inc., Woonsocket, RI, USA). Prior to measurement, the pH meter was calibrated using standard pH 4.0 and 7.0 buffers at 4 °C. Three readings were taken for each sample, and the mean value was calculated.

#### 4.2.2. Shear Force

For shear force evaluation, 200 g of meat was precisely cut into uniform pieces measuring 6 cm × 3 cm × 3 cm. The samples underwent controlled heating in a water bath kettle set at a constant temperature of 80 °C until the core temperature reached 70 °C, as monitored using a digital thermometer. Following this, the samples were allowed to cool to room temperature. Six replicate pieces from each sample were then subjected to shear force measurement using a TA.XT plus texture analyzer (Stable Micro Systems Inc., Godalming, UK), and the mean shear force value was determined.

#### 4.2.3. Water Holding Capacity

The water holding capacity (WHC) was determined using a 25 kg pressure weight with a compression duration of 300 s. The water loss rate was calculated as the percentage of weight loss using the formula: (M_1_ − M_2_)/M_1_ × 100%, where M_1_ represents the initial weight of the meat before compression, and M_2_ represents the weight after compression.

#### 4.2.4. Cooking Loss

Cooking loss was evaluated by subjecting 200 g portions of each sample to heating in a water bath kettle until reaching 80 °C. The percentage cooking loss was calculated based on the weight difference before and after cooking using the formula: (W_1_ − W_2_)/W_1_ × 100%, where W_1_ and W_2_ denote the initial and final weights of the meat, respectively.

#### 4.2.5. Protein Content

The crude protein content of the samples was determined using the Kjeldahl method, following the guidelines outlined in the China National Determination of Protein in Foods Standards (GB 5009.5-2016 [67]). Protein content (X) was calculated using the formula: $X=\frac{(V_{1}-V_{2})\times c\times 0.014}{m\times V_{3}/100}\times F\times 100$, where X represents the protein content of the sample, $V_{1}$ is the volume of the test solution of hydrochloric acid standard titrating solution, $V_{2}$ is the volume of the blank solution of hydrochloric acid standard titrating solution, c is the density of the hydrochloric acid standard titrating solution, m is the mass of the sample, $V_{3}$ is the volume of the digestive solution, and F is the nitrogen to protein conversion factor.

#### 4.2.6. Fat

Total fat was measured using acid hydrolysis, following the guidelines of the China National Determination of Fat in Foods Standards (GB 5009.6-2016 [68]). The fat content was calculated with the formula $X=\frac{m_{1}-m_{0}}{m_{2}}\times 100$. Here, X was the fat content, $m_{1}$ was the mass of fat and the bottle of Soxhlet extractor, $m_{0}$ was the bottle mass, and $m_{2}$ was the sample mass.

#### 4.2.7. Moisture

Water content was determined using the distillation method as described in the China National Determination of Water Content in Foods Standards (GB 5009.3-2016 [69]) followed the formula $X=\frac{m_{1}-m_{2}}{m_{1}-m_{3}}\times 100$. Here, X was the water content, $m_{1}$ was the mass of weighing bottle with sample, $m_{2}$ was the mass of weighing bottle with sample after drying, and $m_{3}$ was the mass of weighing bottle.

### 4.3. RNA Extraction and Sequencing

Total RNA was extracted from the liver and *longissimus dorsi* muscle of Huaxi cattle using the DNF-471-0500 RNA kit (Bositu BIOTECH, Shanghai, China). The integrity of the RNA was determined with a fragment analyzer (Agilent Technologies, Santa Clara, CA, USA) to determine the purity and concentration of RNA samples. RNA samples with RQN ≥ 7 were used to construct libraries. All the sequencing was performed on the DNBSEQ T7 platform (MGI, Wuhan, China).

### 4.4. Transcriptomic Data Analysis

Raw sequencing data were processed using Trimmomatic (v0.39) [70] software to filter out poly-N sequences, splice sequences, and low-quality reads, thereby yielding clean reads. Using the Hisat2 (v2.2.1) [71] software, clean reads were compared to the Bos taurus ARS-UCD1.2 (https://ftp.ensembl.org/pub/release-110/fasta/bos_taurus/dna/ (accessed on 29 July 2023)) reference genome sequences to obtain the location information on the reference genome or gene to obtain mapped reads. Finally, the mapping results were quantified in terms of read counts using the FeatureCounts (v1.5.2) [72] software. Differentially expressed genes (DEGs) between the three groups were identified using Deseq2 (v1.40.2) [73] package in R (v4.3.1), where genes with an adjusted *p*-value < 0.05 and |log_2_FoldChange| > 1 were considered to be DEGs. Notably, genes with mean count < 1 across all samples were excluded from the DEG identification.

### 4.5. Developing the GO Database

To ensure a more comprehensive classification of differentially expressed genes (DEGs) into specific Gene Ontology (GO) categories in subsequent analyses, it was essential to update the background genes in the existing OrgDB database. For this purpose, the makeOrgPackage function of the AnnotationForge package (v1.42.2) in R was used to rebuild the cattle GO database. All GO annotation information was sourced from the EBI GO Annotation program (GOA; www.ebi.ac.uk/GOA/), with the annotation file (ftp.ebi.ac.uk/pub/databases/GO/goa/COW/ (accessed on 16 September 2023)) released on 3 September 2023. In the reconstructed database, background genes for Biological Processes (BP), Cellular Components (CC), and Molecular Functions (MF) were 17,321, 18,343, and 16,327, respectively. This represents an approximate threefold increase in comparison to the original OrgDB database, which contained 4543 genes for BP, 4712 for CC, and 3912 for MF. The updated database was utilized in all subsequent related explorations.

### 4.6. Gene Ontology Enrichment Analysis of DEGs

The GO enrichment analysis for DEGs between each pair of the three groups was conducted using the clusterProfiler (v4.8.3) [74] package in R. We adhered to the default parameters, setting the significance threshold at a *p*-value < 0.05 and a false discovery rate (q-value) <0.05. Pathway clustering was conducted using the EnrichmentMap (v3.3.6) [75] application within Cytoscape (v3.10.1) [76] to construct a network of the identified pathways. This method involved the integration of pathway data. Here, each node in the network represented a distinct pathway, and the edges denoted the shared genes between pathways. Further, the AutoAnnotate (v1.4.1) [77] application was employed to summarize the functional characteristics of the clustered pathways.

### 4.7. Gene Set Enrichment Analysis

Gene set enrichment analysis (GSEA) was performed to further explore the biological significance of gene expression profiles among the three experimental groups. For the GSEA, all genes involved in the pairwise comparisons between the groups were considered. The method involved ranking all genes based on their differential expression levels obtained from the previous analysis. The ranked list of genes was then analyzed against the gene sets from the rebuilt GO database, allowing us to identify whether sets of genes associated with specific BP, CC, or MF were statistically overrepresented at the top or bottom of the ranked list [78]. The analysis was performed using clusterProfiler, where the enrichment score (ES) reflects the degree to which a gene set is overrepresented at the extremes of the entire ranked list of genes. Statistical significance was determined using phenotype-based permutation tests, with the normalized enrichment score (NES) > 1 and adjusted *p*-value < 0.05 considered significant.

### 4.8. Gene Set Variation Analysis

In our study, gene set variation analysis (GSVA) was employed to assess the variation in gene set enrichment across the experimental groups. The first step in this process involved transforming the gene counts into a log_2_ (FPKM + 1), which standardizes the data for comparative analysis. Subsequently, non-parametric estimation of the relative gene set expression levels for the samples was performed using the Gaussian kernel-based cumulative distribution function of the GSVA package to obtain the GSVA score matrix [79]. This step utilized gene sets derived from our previously rebuilt database. Following the generation of the GSVA scores, gene set differential analysis was conducted using the limma (v3.56.2) [80] package. An |logFoldChange| < log_2_1.5 and an adjusted *p*-value < 0.05 were the criteria adopted for selecting significantly differentiated gene sets.

### 4.9. Co-Expression Analysis Based on WGCNA

In this study, we utilized the WGCNA (v1.72-1) [81] package for weighted co-expression gene network analysis (WGCNA). We selected the top 5000 genes with the largest standard deviation from the entire transcriptome data for the next step of analysis. In constructing the scale-free network, we opted for a soft threshold (β) with a fit index (R^2^) of 0.85. Furthermore, the minimum number of genes in a module was set to 30, and the merging threshold for similar modules was set to 0.25. During the analysis, the genes were clustered into different modules, and these modules were subsequently analyzed in association with the three groups: ALF, RF75, and RF50. In each association, we selected the 1–2 most relevant modules as key modules. The genes with the module membership (MM) > 0.8 and gene significance (GS) > 0.2 in these key modules were screened for further analysis.

### 4.10. Screening of Key Genes and PPI Network

We integrated the results of the previous steps of differential expression analysis, GSEA, GSVA, and WGCNA, identifying genes that were significantly expressed across all methods as key genes. Then, protein–protein interaction (PPI) networks for these key genes were constructed using the STRING database (Search Tool for the Retrieval of Interacting Genes/Proteins, https://cn.string-db.org/ (accessed on 22 November 2023)) [82], and the results were visualized using Cytoscape software (v3.10.1). To further confirm the genes associated with beef quality traits, we integrated the key genes with beef quality QTL, which were downloaded from Animal QTLdb (release 51, https://www.animalgenome.org/cgi-bin/QTLdb/index (accessed on 10 December 2023)) [83]. For each key gene identified in this study, we determined its relation with beef quality when it was located within ±100 kb of all QTL of beef quality traits.

### 4.11. Statistical Analyses and Graphs

All statistical analyses were performed in the R (v4.3.1). One-way analysis of variance (ANOVA) with Tukey HSD multiple comparisons were used to compare phenotypic values between groups. *p* < 0.05 was defined as statistically significant. Graphs were generated using several tools. GO pathway network plots were created with Cytoscape and manually adjusted for clarity. Protein–protein interaction (PPI) networks were sourced from the STRING online database. All other figures were generated using R for visualization.

## 5. Conclusions

This study provides valuable insights into the complex interplay among CG, meat quality, and transcriptomic responses in beef cattle. The identification of specific genes and pathways associated with CG under varying degrees of feed restriction offers opportunities for targeted breeding programs aimed at enhancing CG capacity and improving meat quality traits in beef cattle populations. Moreover, our comprehensive approach, integrating phenotypic, transcriptomic, and bioinformatics analyses, sets a precedent for future research investigating CG in livestock production systems. By elucidating the molecular mechanisms underlying CG, our study contributes to the development of strategies to optimize feeding regimes and management practices, ultimately enhancing the efficiency and sustainability of beef cattle production.

## Figures and Tables

**Figure 1 ijms-25-02704-f001:**
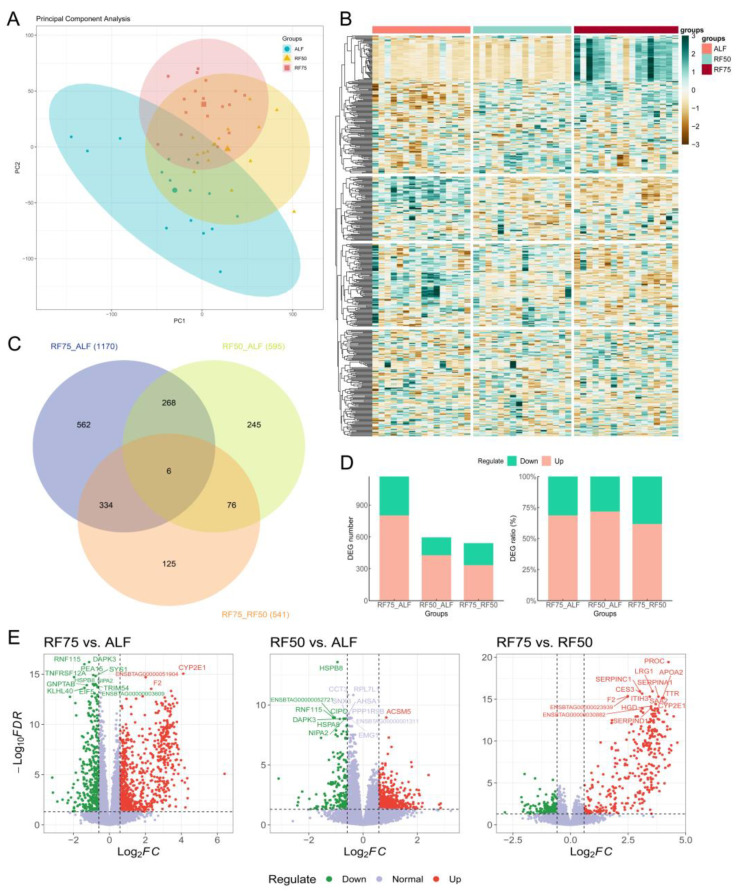
Multifaceted transcriptomic analysis of compensatory growth in Huaxi cattle under different feeding restrictions. (**A**) Principal component analysis (PCA) depicting the distribution of samples among three experimental groups. (**B**) Heatmap of the top 1000 genes with the highest standard deviation in expression across all groups. (**C**) Venn diagram illustrating the overlap of differentially expressed genes (DEGs) among the three comparative groups, highlighting shared and unique gene expression changes. (**D**) Bar plots showing the number of DEGs (**left**) and the relative proportion of up- and down-regulated DEGs (**right**) identified in the specified comparative groups. (**E**) Volcano plots representing the distribution of DEGs in the RF75 vs. ALF (**left**), RF50 vs. ALF (**middle**), and RF75 vs. RF50 (**right**) comparative groups, with points colored to indicate up-regulation (red), down-regulation (green), or no significant change (grey-blue).

**Figure 2 ijms-25-02704-f002:**
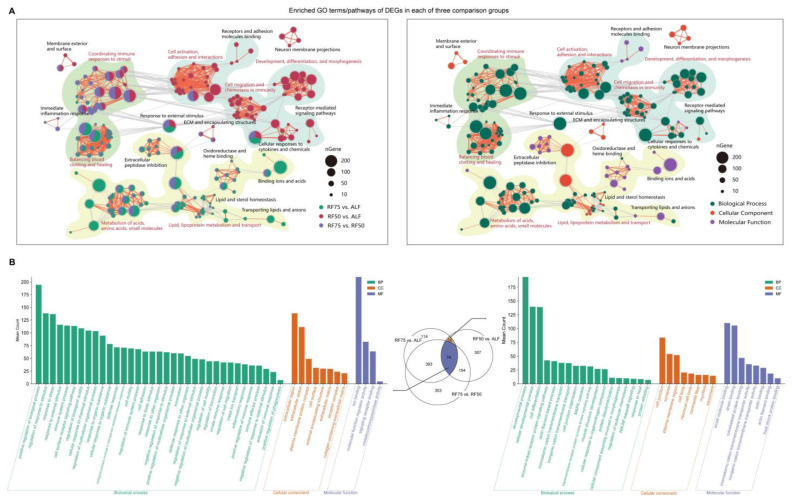
Functional enrichment analysis of DEGs in compensatory growth. (**A**) Network plots constructed from the top 100 GO terms identified in each of the three comparison groups. Each node represents a pathway, with the pie size indicating the gene number associated with the GO term, the thickness of the edges representing the number of shared genes, and the colors denoting different comparison groups (**left**) or ontology categories (**right**). (**B**) Bar charts illustrating the number of GO pathways shared across all three comparison groups (**left**) and shared between the RF75 vs. ALF and RF50 vs. ALF comparison groups (**right**).

**Figure 3 ijms-25-02704-f003:**
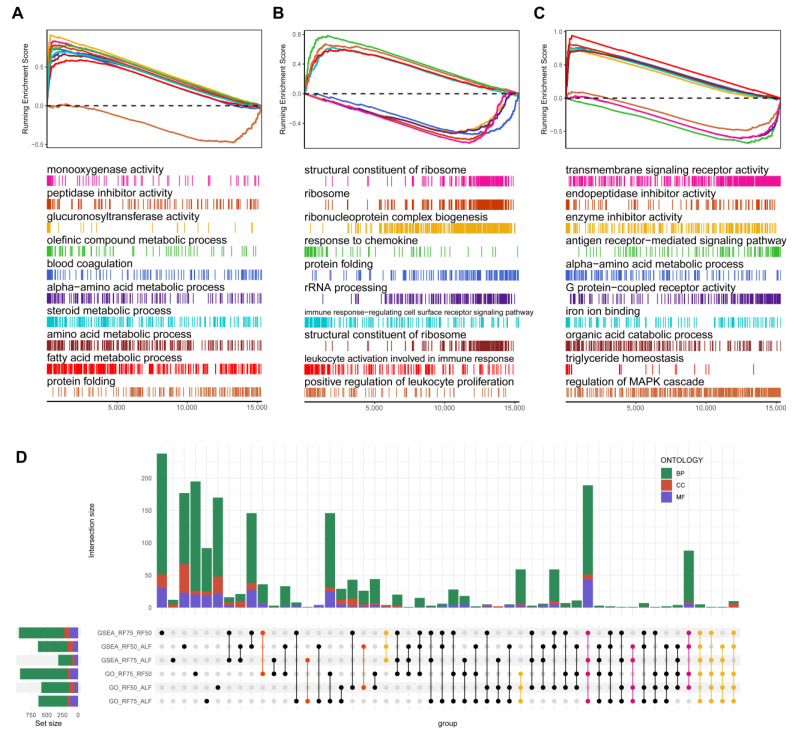
The top 10 representative pathways from GSEA results comparing (**A**) RF75 vs. ALF, (**B**) RF50 vs. ALF, and (**C**) RF75 vs. RF50 groups. Each broken line represents a pathway with its enrichment score plotted along the *x*-axis, and each vertical line represents a gene contributing to the enrichment score of the pathway. (**D**) UpSet plot combining results from GSEA and differential expression GO enrichment analyses, showing the intersections of significantly enriched pathways and gene ontologies across the different comparative groups. The size of each set is indicated by bar height, while the intersections are represented by connected dots, color-coded by ontology category (BP: biological process, CC: cellular component, MF: molecular function).

**Figure 4 ijms-25-02704-f004:**
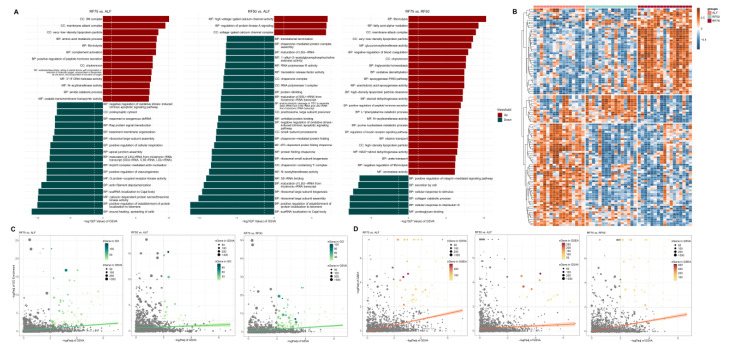
Gene set variation analysis (GSVA) of compensatory growth. (**A**) The top 30 significantly gene sets identified from the three comparisons. Red and green indicate up-regulated or down-regulated pathways, respectively. (**B**) Heatmap illustrating the correlation between all significant gene sets and sample groups. Scatter plots representing the intersection of GSVA and (**C**) GO enrichment analysis or (**D**) GSEA, with axes denoting the −logPadj from each analysis and point size and color intensity reflect the number of gene sets.

**Figure 5 ijms-25-02704-f005:**
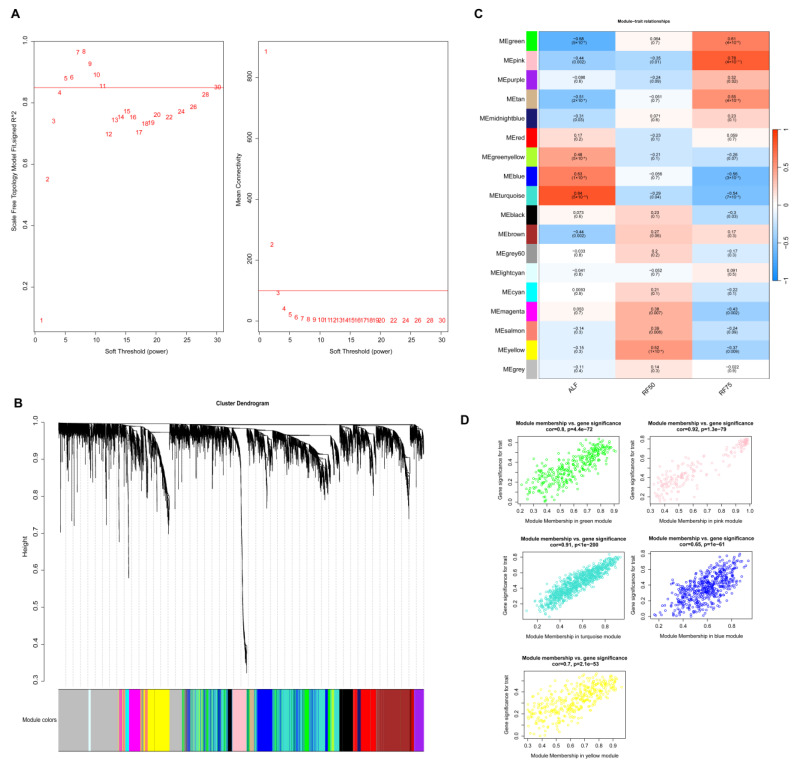
Weighted gene co-expression network analysis. (**A**) Analysis of network topology for various soft-thresholding powers. (**B**) Clustering dendrogram of genes, with the color row underneath the dendrogram displaying the module assignment determined using the dynamic tree cut method. (**C**) Relationship between gene modules and different feeding treatment groups. The value outside the parentheses expresses Pearson’s correlation coefficients between the module eigengenes and groups, and the number within the parentheses is the *p*-value. (**D**) Scatterplots of gene significance (GS) for external traits vs. module membership (MM) in selected modules.

**Figure 6 ijms-25-02704-f006:**
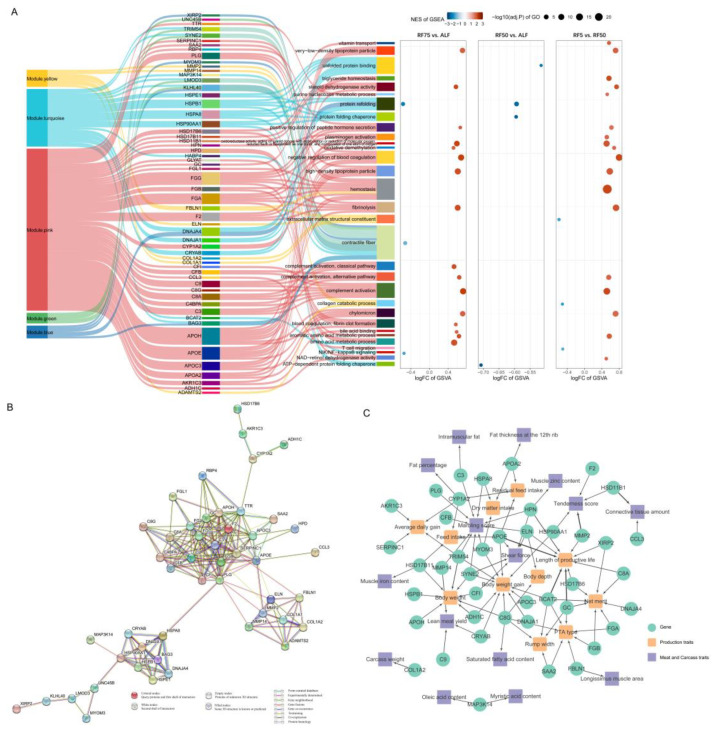
Integrated analysis to identify key genes related to compensatory growth. (**A**) Sankey–bubble plot of key genes and pathways. The Sankey plot demonstrates integrated flow between modules, genes, and pathways, where the size of each bubble corresponds to the Gene Ontology (GO) enrichment analysis’s −log_10_(adj.*p*), the *x*-axis represents the logFC from GSVA, and the bubble color indicates the normalized enrichment score (NES) from GSEA. (**B**) Network of protein–protein interaction (PPI) among key genes. (**C**) Relationships of key gene and production or meat and carcass traits quantitative trait loci (QTL). Cyan circles: key genes. Orange squares: QTL of production traits. Purple squares: QTL of meat and carcass traits.

**Table 1 ijms-25-02704-t001:** Effect of different feeding treatments on cattle growth performance.

Trait/Variable	ALF	RF75	RF50	SED	P_ANOVA_
Body weight (BW), kg					
Start, d0	375.76	379.41	375.74	42.95	0.991
Mid of RF_d28	415.24 ^Aa1^	391.43 ^a^	358.52 ^b^	55.16	0.001
End of RF, d49	447.46 ^A^	401.43 ^B^	363.52 ^C^	50.46	<0.001
Realimentation_d135	538.33 ^Aa^	529.37 ^a^	495.07 ^Bb^	50.35	0.007
Realimentation_d165	577.89 ^Aa^	572.32 ^a^	537.96 ^Bb^	50.06	0.01
Slaughter, d380	767.18	774.05	759.9	59.00	0.751
Average daily weight gain (ADG), kg/d					
RF_d0–28	1.52 ^A^	0.52 ^B^	−0.61 ^C^	1.01	<0.001
RF_d28–49	1.4 ^A^	0.43 ^B^	0.22 ^B^	0.79	<0.001
RF, d0–49	1.46 ^A^	0.45 ^B^	−0.25 ^B^	0.78	<0.001
Realimentation_d49_135	1.04 ^B^	1.47 ^A^	1.51 ^A^	0.31	<0.001
Realimentation_d135_165	1.32	1.46	1.43	0.45	0.552
Realimentation_d165_380	0.82	0.93	0.96	0.28	0.196
Realimentation, d49–380	0.93 ^B^	1.12 ^A^	1.15 ^A^	0.21	<0.001
Entire Period	1.00	1.04	0.97	0.17	0.45

^1^ Different lowercase letters indicate significant differences with *p* < 0.05, different uppercase letters indicate significant differences with *p* < 0.01.

**Table 2 ijms-25-02704-t002:** Effect of different feeding treatments on carcass and meat quality traits of cattle.

Trait/Variable	ALF	RF75	RF50	SED	P_ANOVA_
Carcass trait					
Carcass Weight, kg	414.2	419.36	404.00	31.98	0.30
Striploin ^1^, kg	6.40	6.41	6.23	0.76	0.70
High-rib, kg	11.60	12.00	11.69	1.47	0.66
Tenderloin, kg	4.12	4.10	3.99	0.51	0.68
EMA ^3^	104.6	108.71	106.25	11.16	0.5
Meat Quality					
pH_24_	5.45 ^Aa2^	5.80 ^Bb^	5.65 ^b^	0.29	<0.001
Shear Force	9.21	9.01	9.19	2.70	0.966
WHC	0.38	0.36	0.37	0.04	0.208
Cooking Loss	0.3 ^b^	0.29 ^ab^	0.24 ^a^	0.06	0.029
Meat Composition					
Protein, %	23.55 ^a^	22.77 ^b^	23.26 ^ab^	0.78	0.014
Fat, %	2.98 ^a^	3.8 ^b^	3.34 ^ab^	0.93	0.042
Moisture, %	72.89	72.06	72.33	1.22	0.146

^1^ All weights recorded of meat pieces in the table are from left half carcasses. ^2^ Different lowercase letters indicate significant differences with *p* < 0.05, and different uppercase letters indicate significant differences with *p* < 0.01. ^3^ EMA = eye muscle area; WHC = water holding capacity.

## Data Availability

The datasets are available upon request from National Centre of Beef Cattle Genetic Evaluation, Beijing, China (Email: pingguzhongxin@126.com).

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
