# Peer review of "Transcriptome Analysis of Compensatory Growth and Meat Quality Alteration after Varied Restricted Feeding Conditions in Beef Cattle"

_ijms, 2024, doi:10.3390/ijms25052704_

Round 1

Reviewer 1 Report

Comments and Suggestions for Authors

Did you check the food intake for animals after feed restriction? 

Is it necessary to wait a year to collect the samples? Each stage the gene expression might be different. Please clarify why do you think the information from one year old animal is critical. Data from different stages should be more helpful to understand the effect of CG. 

Discuss more about how can you apply the information you got based on your bioinformatics analysis to practice or in the future study. I am not surprised that you can see some difference but what do these finding suggest and what is the implication? This should be the key of your study. 

Author Response

Comments 1: Did you check the food intake for animals after feed restriction?

Response 1: Thank you for your insightful question. After the feed restriction phase, we conducted short-term monitoring of food intake, alongside collection of fecal and urinary outputs to assess the metabolic processing of nutrients and energy. This data is part of a comprehensive analysis planned for a separate, yet unpublished study focused on animal nutrition, which explains its absence from the current manuscript.

Comments 2: Is it necessary to wait a year to collect the samples? Each stage the gene expression might be different. Please clarify why do you think the information from one year old animal is critical. Data from different stages should be more helpful to understand the effect of CG.

Response 2: We appreciate your query on the timeline for sample collection. Our decision to wait a year before collecting samples was informed by existing research primarily focused on the initial phases of growth compensation. Given that much is already understood about the early effects of CG, our study aims to shed light on the long-term or ultimate impacts of CG on adult animals, particularly in the context of the beef industry's production cycle. This long-term perspective is crucial for understanding the mechanisms through which CG operates. However, acknowledging your suggestion, we are considering incorporating additional time points in future research to provide a more comprehensive understanding of CG's effects across different developmental stages.

Comments 3: Discuss more about how can you apply the information you got based on your bioinformatics analysis to practice or in the future study. I am not surprised that you can see some difference but what do these finding suggest and what is the implication? This should be the key of your study.

Response 3: We thank you for highlighting the importance of discussing the implications of our findings. Indeed, the observed differences raise critical questions about the role of CG in animal growth and development. Our study suggests potential pathways through which CG might exert its effects, which could have far-reaching implications for animal husbandry and genetic studies. In response to your suggestion, I have enhanced the discussion section and reorganized the conclusion to more effectively address the implications of our research. The refined discussion provides deeper insights into how our findings contribute to the existing body of knowledge and their potential practical applications. We believe these updates strengthen the manuscript and address the key focus of our study.

We believe these revisions have addressed the concerns raised by the reviewer and have significantly improved the manuscript. We are thankful for the insightful feedback and the opportunity to enhance our work. We look forward to any further suggestions.

Reviewer 2 Report

Comments and Suggestions for Authors

The manuscript treats an important issue for both animal science and meat science.  The study adopts multidisciplinary approach to achieve the aim that is generally correct.

The introduction present the necessary information for the background of the study. The aim is also very clearly defined. 

The material and method section is described in details, however, the analysis concerning the meat quality characteristics (pH measurement, shear force, cooking loss, WHC, proximate composition)  are not described. Yet, they are presented in a separate table have significant importance and should be included in the description of the materials and  methods.

The number of the animals per group is presumably 20, however this is nowhere assigned. Here is my main concern about the soundness of the experimental design. 20 animals is a low number for performing genetic analyses. I am quite aware that there are a number of studies reporting genetic analysis is  even lower number of animals, mainly large ruminants, however, in my opinion for sound conclusions concerning genetic traits we need at least 50 animals.

The study results are very well illustrated in graphic material. Please, provide the software used to create the graphics.

The results are well presented. Some specific remarks: please avoid the term "highly significant". The differences are either significant or not. In this regard, carefully check the footnotes below the tables: for both tables the significance is presented in uppercase and lowercase letters to present the level of significance. However, it is stated that the significance is presented in lowercase for both p<0.05 and p<0.01.

The discussion should be more profound in the part concerning the physical parameters of the meat. It is interesting why there is no instrumental colour measurements. They are closely connected to pH and WHC. The discussion of pH is not appropriate, as well as the citing reference [6]. The lower pH values usually does not meat slower glycogen breakdown, but accumulation of lactic acid due to breakdown of glycogen. Furthermore, cooking loss depends on multiple factors, and importance of breed is overestimated here. So, please, discuss this parameter in the light of your study and the factor that you study- restricted feeding.

The results are generally supported by the results, however sound conclusions need larger sample of animals.

Author Response

Comments 1: The material and method section is described in details, however, the analysis concerning the meat quality characteristics (pH measurement, shear force, cooking loss, WHC, proximate composition) are not described. Yet, they are presented in a separate table have significant importance and should be included in the description of the materials and methods.

Response 1: We acknowledge the omission in the detailed description of meat quality characteristics analyses in the Materials and Methods section. We have now revised this section to include detailed methodologies for pH measurement, shear force, cooking loss, water holding capacity (WHC), and proximate composition. These changes ensure that the experimental procedures are transparent and reproducible.

Comments 2: The number of the animals per group is presumably 20, however this is nowhere assigned. Here is my main concern about the soundness of the experimental design. 20 animals is a low number for performing genetic analyses. I am quite aware that there are a number of studies reporting genetic analysis is even lower number of animals, mainly large ruminants, however, in my opinion for sound conclusions concerning genetic traits we need at least 50 animals.

Response 2: We appreciate the opportunity to clarify the rationale behind the selection of 20 animals per group in our study. This decision was guided by the objective to ensure the comparative genomics analysis was conducted within a controlled and homogenous group. The animals selected for our study were primarily from the same farm, of the same growth stage, and closely related as half-siblings. This strict selection criteria, aimed at minimizing variability due to external factors, inherently limited the availability of suitable animals. Initially, our design included 75 animals (25 per group), but to maintain control over variables such as initial body weight and age, we adjusted the number to 60 animals in total. Additionally, logistical constraints related to the housing and management of large ruminants (e.g., space availability, environmental conditions, and individual feeding systems) made the handling of a larger cohort impractical at this stage, potentially introducing systemic and random errors. We acknowledge the reviewer's concern regarding the sample size for genetic analysis and have elaborated on these considerations in the manuscript to provide a comprehensive understanding of our experimental design's context and limitations. We aim to explore larger cohorts in future studies as we advance our research infrastructure.

Comments 3: The study results are very well illustrated in graphic material. Please, provide the software used to create the graphics.

Response 3: We have updated the manuscript to include the software used for creating graphical materials.

Comments 4: The results are well presented. Some specific remarks: please avoid the term "highly significant". The differences are either significant or not. In this regard, carefully check the footnotes below the tables: for both tables the significance is presented in uppercase and lowercase letters to present the level of significance. However, it is stated that the significance is presented in lowercase for both p<0.05 and p<0.01.

Response 4: We appreciate the reviewer's point on the use of "highly significant." We have revised our manuscript to ensure that the presentation of statistical results is precise and standardized, removing any instances of "highly significant" and ensuring that the significance levels are correctly stated in both the text and table footnotes as per standard statistical reporting practices.

Comments 5: The discussion should be more profound in the part concerning the physical parameters of the meat. It is interesting why there is no instrumental colour measurements. They are closely connected to pH and WHC. The discussion of pH is not appropriate, as well as the citing reference [6]. The lower pH values usually does not meat slower glycogen breakdown, but accumulation of lactic acid due to breakdown of glycogen. Furthermore, cooking loss depends on multiple factors, and importance of breed is overestimated here. So, please, discuss this parameter in the light of your study and the factor that you study- restricted feeding.

Response 5: We have expanded the discussion on the physical parameters of meat, including a more detailed analysis of pH, cooking loss, and their implications for meat quality in the context of restricted feeding. Regarding the measurement of meat color, we encountered an unforeseen challenge during the carcass processing phase – our meat colorimeter was damaged, which necessitated the use of a color comparison method as an alternative. Aware of the higher error margin associated with color comparison methods, as well as the absence of the expected significance after the analysis, we initially refrained from including meat color data in our analysis. The revised discussion now includes a corrected interpretation of pH and their impact on meat quality, as well as a more balanced view on the influence of breed and other factors on cooking loss.

Comments 6: The results are generally supported by the results, however sound conclusions need larger sample of animals.

Response 6: We recognize the reviewer's concern regarding the sample size for drawing sound conclusions on genetic traits. In the revised manuscript, we further discuss the implications of our sample size on the study's conclusions and how future research could build on our findings by incorporating larger animal cohorts. This reflection is aimed at providing a balanced view of our study's contributions and limitations.

We believe these revisions have addressed the concerns raised by the reviewer and have significantly improved the manuscript. We are thankful for the insightful feedback and the opportunity to enhance our work. We look forward to any further suggestions.

Round 2

Reviewer 2 Report

Comments and Suggestions for Authors

The authors have provided thorough and sound explanations of the remarks. I agree with the revised version and elaborations in particular concerning the small sample size. I recommend to proceed to publication.